# 'Hybrid' top down bottom up health system innovation in rural China: A qualitative analysis

Joris van de Klundert[1,2]*, Dirk de Korne[2,3], Shasha Yuan[4], Fang Wang[4], Jeroen van Wijngaarden[2]

**1** Prince Mohammad Bin Salman College of Business & Entrepreneurship, KAEC, Kingdom of Saudi Arabia, **2** Erasmus School of Health Policy and Management, Erasmus University Rotterdam, Rotterdam, Netherlands, **3** Health Services Innovation, University of Tasmania, Hobart, Australia, **4** Institute of Medical Information and Library, Chinese Academy of Medical Sciences & Peking Union Medical College, Beijing, China

\* jklundert@mbsc.edu.sa

**Data Availability Statement:** The interview data (interview data with individuals and focus groups) are all collected by the researchers. The World Bank stores the data. The authors can only share

## Abstract

### Introduction

China has made considerable progress with health system reforms in recent years. Rural China, however, has lagged behind as the diversity of needs of China's 3,000 rural counties were not always well addressed by national top-down reforms. China's Rural Health Reform Project Health XI (HXI) piloted a *hybrid* process of top down and bottom up implementation of health system reforms which were tailored to rural county level needs and covered a population of more than 21 million. Different studies provide evidence that HXI counties have achieved substantial benefits given the relatively limited investment. The Effectiveness of HXI subsequently raises the question *how* the hybrid approach may have resulted in effective implementation of interventions. We answer this question to advance understanding of hybrid approaches in general and in the rural Chinese context in particular, where the bottom-up elements might match poorly with the traditional organisational culture and learning style.

### Materials & methods

We conducted an in-depth qualitative analysis in three 'best practice' counties, performing document-analyses, observations, semi-structured individual and group interviews. In alignment with the research question, this study is of an explorative nature and follows a sequence of deductive and inductive steps

### Results

HXI struggled initially as counties had difficulties to take initiative and autonomously select and adapt their own reforms. The initial reforms required multiple improvement iterations before achieving the planned results. The effectiveness of these bottom up reform processes has been aided by tight top down supervision and extensive domestic expert involvement. County level leadership is seen as essential to align the top down and bottom

these qualitative data with their permission. Additional data, in the form of documents and reports can be shared freely. The authors didn't have any privileged access to data that others would not have. Interested researchers can contact Dr. Shuo Zhang from the World Bank via email (szhang2@worldbank.org) in regards to data access.

**Funding:** The external evaluation of HXI was commissioned by the World Bank and funded by The World Bank, IDF Grant for Capacity Building of Evaluation of China's Health System Reform Pilot (No. TF013943). The originally collected data are therefore owned by the Word Bank and can only be shared by the authors with consent of the World Bank. The funders had no role in study design, data collection and analysis, decision to publish, or preparation of the manuscript.

**Competing interests:** The authors have declared that no competing interests exist.

up structures and processes. Where successful, HXI has changed mind-sets and counties developed generic health improvement capabilities.

## Conclusion

Tailoring innovations to fit local needs formed a severe challenge for the three 'best practice' counties studied. A 'change of mindset' to actively take initiative and assume autonomy was needed to advance. Top down supervision and extensive support of experts was required to overcome the barriers. The studied counties finally achieved sustainable improvements and developed double loop learning capabilities beyond HXI objectives. Taken together, the above findings suggest that the continuum of healthcare reform implementation approaches in which hybrid approaches reside—from bottom up to top down—has two dimensions: a content dimension and a procedural dimension. Enabled by top down procedures, counties were able to bottom up tailor the content of best practice innovations to fit local needs.

## Introduction

Since the start of the millennium, the Chinese government has implemented a series of national health system reforms, which have achieved considerable improvements [1, 2]. Within this series, 2009 formed a watershed year in which a comprehensive round of reforms consisting of five pillars was announced with the aims of '1) expanding the coverage of and benefits provided by subsidised health insurance; 2) implementing a drug reform scheme with zero mark-up on listed essential medicines; 3) improving access to and the quality of primary health care; 4) implementing basic public health screening and management at the community level for all; and 5) implementing public hospital reform' [3]. These reforms have been further strengthened by additional health policy measures specifically targeted at China's 800 million rural citizens, who formed more than half of China's population in 2009 and comprised close to 10% of the global population [4–8]. These efforts have predominantly followed a *'top-down'* approach, i.e., 'a policy-centred and rational process that begins with an authoritative decision by senior policy officials' [9]. Their implementation has followed a 'planned, regulated, properly managed' process in which interventions were disseminated or cascaded from the national level to be adopted at the local level [10, 11].

Despite the accomplishments of China's health reform by top-down approaches, the World Bank, the British Department of Foreign Investment (DFID), and the Chinese government (i.e., The National Health & Family Planning Commission (NHFPC), the National Development & Reform Commission, and the Ministry of Finance) agreed in 2008 that the health services for the rural population were lagging behind and that their wide variety of contexts, populations, and needs had not always been served equitably by the implemented top-down reforms [4]. Therefore, these stakeholders jointly set up China's Rural Health Reform Project Health XI (HXI) [4]. The design and intention of HXI was to pilot the implementation of health system reform interventions in 40 rural counties from eight rural provinces following a *'bottom-up'* approach [4, 12–14]. This approach intended each county to implement reform interventions tailored to its specific needs and context. HXI took a very specific bottom-up approach in which counties were required to follow a pre-structured process of selection and adaptation of international best practices and to target a predefined set of reform areas and objectives [4, 12–14]. To this purpose, HXI employed an extensive formal planning and control structure, as contractually agreed to by The World Bank, DFID, and China's national

government, and complemented 'bottom-up' elements with 'top-down' elements. This somewhat contrasts with a more common definition of bottom-up reform, which refers to developing policy and interventions in 'a joint process of formulation and implementation involving networks and coalitions' [9]. In this study, we analyse HXI's hybrid bottom-up/top-down approach to the implementation of health reforms.

Project HXI lasted from December 2008 to November 2014. S1 and S2 Appendices provide detailed information about the 22 indicators representing the project objectives and the participating counties and provinces. The project started less than half a year before the announcement of the National Health Reform of 2009 [3] and targeted reforms in three areas that were well aligned with the National Health Reform: 1) improving rural health financing; 2) improving quality, efficiency and cost control in service delivery; and 3) financing and organizing core public health functions [4, 12]. Examples of selected and adapted interventions for each of the three areas are 1) improving NCMS coverage and integration with MA (to improve rural health financing), 2) introducing new hospital payment systems (e.g., case-based payment) and introducing performance-based payment in hospitals and lower-level facilities, and 3) training public health workers as well as the population on hypertension and diabetes management. See Table 1 of [15] for a comprehensive overview.

A quantitative difference in difference study provides evidence that, in comparison to carefully selected rural control counties, HXI counties have achieved 'substantial benefits given the relatively limited investment' [15]. These benefits include 'significant decreases in outpatient expenditure and financial strain, and improvements in public health services provision, as well as in two out of five dimensions of the EQ5D model for health-related quality of life (mobility, and pain/discomfort)' [15]. Evidence of these results is further strengthened by evaluation studies of specific interventions [14, 16, 17]. The official evaluation by the World Bank reports that, with one exception, all performance targets on the 22 project indicators were either 'achieved' or 'surpassed' [4]. The effectiveness of HXI subsequently raises an interest in *how* the hybrid approach may have resulted in effective implementation of interventions at the county level, where previous top-down approaches were perceived to have been less effective.

The exploration of the workings of the HXI approach is of particular interest because of its rural Chinese context. The scant existing evidence on hybrid approaches is typically derived from Western studies and may lack validity in rural China health systems [11, 18]. For instance, county-level stakeholders may find it difficult to operate in a bottom-up manner, as China's organizational culture is known to be hierarchical, have a 'large power distance', and rely on top-down decision making [19, 20]. Likewise, bottom-up approaches presuppose autonomy, an active attitude, and initiative, which are characteristics that are typically not associated with large power distance [21]. Rather than following a creative, explorative, process, the Chinese rural culture characteristically prefers receiving instructions and seeking external expertise [22]. Moreover, the cultural values of rural counties typically emphasize conformity and harmony and consequently value evolutionary, incremental innovations over revolutionary and disruptive forms [22, 23]. Counties may therefore struggle to identify and implement bottom-up reforms and innovations that are 'challenging to the status quo, authority and traditions', as implied by the ambitious HXI objectives [4, 23].

While the present scientific understanding of hybrid approaches is limited, the results of HXI suggest that they may have considerable relevance for the rural Chinese health system context. The bottom-up elements appear to enable the effective tailoring of reform to the variety of local health needs of rural counties, thus overcoming previously identified shortcomings of top-down approaches [4]. Hence, we set out to explore how HXI brought about its results, aiming to advance the understanding of hybrid approaches in general and in rural China in particular. The research question is as follows: 'How has the effective implementation of health

reform interventions occurred in the hybrid bottom-up/top-down approach taken by HXI in the context of rural China?'

## Materials & methods

### Design and reporting

In alignment with the research question, this study is of an explorative nature and follows a matching sequence of inductive and deductive steps [24]. Based on an initial exploration of our cases (inductive), we identified relevant (guiding) concepts from the literature to further sharpen the research aims, research question and methods, and specifically the interview questions (deductive). During the analyses, the data were first categorized based on the guiding concepts (deductive). Subsequently, an inductive approach was used to analyse the data and identify relevant themes [24–26].

The methods and reporting follow the SRQR protocol for reporting qualitative research (which encompasses the COREQ guidelines [27, 28]). The full SRQR checklist is provided in S3 Appendix. Data were collected using document analyses, observations, individual interviews and group interviews.

### Initial data collection and guiding concepts

The initial activities consisted of a week of field visits in March 2012 and the analysis of all the project documentation available up to and including the formal mid-term evaluation of HXI in 2011 [12, 13]. This initial analysis revealed that HXI used a hybrid blend of top-down and bottom-up elements to spread health reform innovations. These initial findings formed the basis for the conceptual perspective of exploring how a hybrid approach occurred in the rural Chinese health reform pilots of HXI.

Based on the first exploration, other concepts that were hypothesized to be of relevance for the rural Chinese context were also identified in consultation with the scientific literature. The first of these concepts relates to the working of bottom-up processes in an organizational culture with high power distance [19–21]. The second concept relates to the Chinese, i.e., Confucian, view of learning by means of knowledge transfer, as opposed to discovering knowledge by experimentation [22, 23]. These concepts were therefore specifically addressed in the data collection.

The initial analysis further revealed that the World Bank provided and taught a model of health reform stages to support counties in their implementation processes [4]. This staged model closely resembled the well-known evidence-based framework for the spread of health service innovations in Greenhalgh [10, 11]. By merging the terms of the World Bank model into the evidence-based framework, the model depicted in Fig 1 arose. This merged model, which emphasizes the staged and cyclic nature of health system reform, supported the interviewing process, as it enabled us to collect data following an evidence-based framework that completely captures the stages of implementation of innovations using terminology that HXI participants were already familiarized with. The model contains a subcycle formed by the stages of adaptation, implementation, and evaluation, recognizing that various cyclic iterations might be conducted per intervention.

After these initial activities and the first inductive and deductive analysis steps, the data collection continued as detailed below at the project's end in 2014.

### Data collection methods

The subsequent qualitative research addressing how the effective implementation of health reform interventions occurred primarily relied on interviews and focus group discussions.

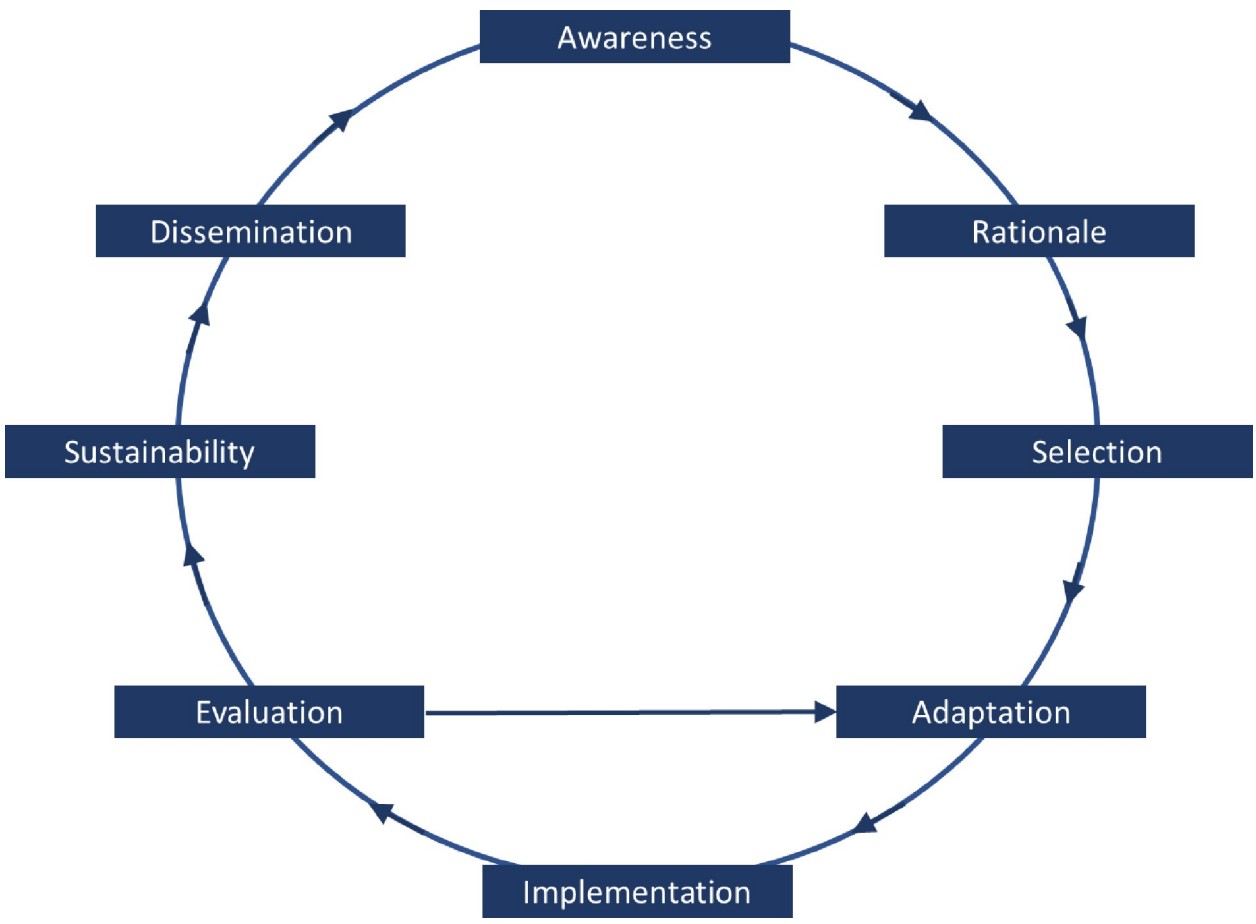

**Fig 1. The staged implementation cycle.**

These interviews took place at the HXI project level (NHFPC, World Bank, DFID, international and national experts involved) and in three participating counties: Mei Xian in Shaanxi Province, Jiulongpo in Chongqing Province, and Xi Xian in Henan Province. Each of the interviews and focus group discussions was organized according to the staged implementation cycle of Fig 1 to ensure that the reform implementation processes considered were covered completely. Moreover, the topics of bottom-up and top-down mechanisms were explicitly addressed, as were the practices of learning. The interview protocol can be found in S4 Appendix. These interview data were further complemented by an analysis of all formal internal and external HXI documents (see S5 Appendix).

## Selection of counties and respondents

The three counties were selected by the researchers from three lists of counties that were considered best-practice counties by the HXI project management office CPSM. Each of these three lists included best-practice counties for one of the three reform components of HXI. Xi Xian was selected for its innovative hospital financing reform, Mei Xian was selected for its comprehensive service delivery management reform, and Jiulongpo was selected for its achievements in public health improvement. Moreover, the three counties together covered the included provinces relatively well from a geographical and economic perspective. The method of selecting only from the supposedly effective counties does not allow a full evaluation

of HXI; it does, however, suit the research objective of exploring how the hybrid approach of HXI worked in those counties where it has been effective.

Respondents were selected from the international parties involved in the health reforms and from multiple levels within China, i.e., the national level, the provincial level, the county level, the organizational level, and the professional level. Moreover, the researchers conducted group interviews with patients. In addition, international-, national-, provincial-, and county-level experts who were involved were selected for interviews. Respondents from the World Bank, DFID, and the National Health and Family Planning Commission were selected and contacted directly by the researchers for interviews. The CPSM, in collaboration with provincial and county health bureau administrators, selected and invited for interviews the respondents indicated by the researchers at the local level. Likewise, the CPSM selected experts for interviews and provided logistic support. Table 1 gives a detailed account of all the respondents.

All the interviews were conducted face-to-face, except for the interviews with one World Bank official, a DFID representative, and a formerly involved CPSM leader, which took place between July and September 2014 via telephone/Skype. The phone/Skype interviews were conducted in English by JvdK.

The face-to-face interviews with the Chinese respondents were always conducted by a pair of researchers, one of whom was of Chinese nationality (FW or SY) and one of whom was from abroad (JvdK or DdK). These interviews were conducted as follows: (1) The researchers' questions were always asked in English; (2) A certified interpreter translated the questions into Chinese and then asked them to the respondents; (3) The respondents answered in Chinese; (4) The certified interpreter then translated the responses from Chinese to English; and (5) The next question was asked, etc. On two occasions, the interview was moderated by SY instead of a certified interpreter.

Given the size of HXI, which encompassed 40 counties in eight provinces in which more than 200 innovations have been implemented and which covered a population of more than 20 million citizens [15], completeness of data collection or analysis, for instance, as expressed through various forms of data saturation criteria [27], is hard to obtain. This problem persists when restricting the research design to three counties. In combination with the practical delays

**Table 1. Respondents by level and location.**

| Level | Experts | Shaanxi | Chongqing | Henan |
|---|---|---|---|---|
| Worldbank | 2 | | | |
| DFID | 1 | | | |
| CPMS | 4 | | | |
| National experts | 3 | | | |
| Provincial level experts | 2 | | | |
| County level experts | 1 | | | |
| Provincial HB | | 1 | 2 | 1 |
| County HB | | 3 | 4 | 2 |
| County H | | - | - | 2 |
| Health professional THC | | - | - | 2 |
| THC management | | 3 | 3 | 1 |
| Health professional VC | | 2 | 3 | 2 |
| Patients (FDG) | | 3 (1) | 10 (2) | 5 (1) |
| Total | 13 | 12 | 22 | 15 |

between data collection, transcription, translation and analysis, a classical criterion of data saturation was therefore difficult to adopt. Instead, we relied on alternatives that matched our research approach, which combined inductive and deductive steps. From the broader perspective of completeness, let us first note that we sampled counties from each of the three reform areas of HXI and respondents from all levels of HXI, including patients. As much as possible (e.g., due to obtaining consent from respondents), we interviewed multiple respondents of each relevant level and type. For some levels, only one respondent existed. Our interview protocol and coding based on guiding concepts ensured a priori thematic saturation of the deductive analysis steps [29]. The open coding and joint iterative analyses by DdK and JvW in the inductive stages converged towards inductive thematic saturation [29].

Each interview lasted between one and three hours. All the respondents provided their explicit consent to be interviewed verbally and on record, as well as for their interviews to be recorded for the purpose of anonymous data collection, after having accepted the invitation to participate and being made aware that the data collection was part of the external evaluation of HXI commissioned by the World Bank. The interviews never discussed patient cases or individual health-related information. Both project HXI itself and the evaluation study were approved by the World Bank and the National Health & Family Planning Commission.

Each of the three county visits in the project counties started with a general opening meeting, after which three days of site visits and interviews followed. Each of the county visits was concluded with a group interview with all respondents in which the findings were discussed and cross-checked.

## Data analysis

The English texts of the interviews were transcribed using audio files of the full interviews. The Chinese audio content was used to resolve unclarities in the English text when transcribing. Relevant Chinese documents were translated to English as well. The interview and document data were sequentially coded and categorized using labels from our guiding concepts (deductive) and open labels (inductive) developed by DdK and JvW [24]. The labels were used to identify recurrent themes [24–26]. We tabulated the frequency of different topics and identified and compared the respondents' views on the implementation of interventions in HXI.

Based on the interview protocol, including the framework presented in Fig 1 and the concepts of top-down versus bottom-up reform and learning practices [10–11, 19–23], a detailed analysis of the data in each of the three counties resulted in three case study reports composed by DdK, JvW and JvdK [20]. These analyses used triangulated data from county-level respondents, national and international respondents, and document analysis. The insightful brief case reports in S6 Appendix are derived from the case study reports. Pattern matching was used to synthesize the empirical findings from the three case studies and the project level data analysis (DdK, JvW, JvdK). To further increase construct validity, Chinese and Dutch expert collaborators and CPSM staff reviewed the project reports and synthesis [14]. The resulting synthesis is presented below. The funders had no role in the study design, the data collection and analysis, the decision to publish, or the preparation of the manuscript.

## Results

### A difficult start

HXI began in 2008, intending to initiate reform immediately in each of the forty participating counties. For this purpose, each county was asked to set their own reform priorities using baseline measurements based on the findings of the National Household Survey [30, 31] and the set of 22 indicators that was especially designed for HXI (see S1 Appendix). The counties were

subsequently expected to select, adapt, and implement health system innovations (sometimes referred to as interventions below) according to yearly plans developed at the county level to address the priorities they formulated themselves as being necessary to meet their specific health needs. These priorities needed to cover the 22 HXI indicators and might have addressed additional performance measures as well. Each county was subsequently required to report measurable progress in each of the three HXI areas: 1) health system financing, 2) service delivery, and 3) public health.

Over the entire duration of HXI, the CPSM arranged feedback, organized conferences and other events, facilitated access to expert support, and monitored progress. Moreover, HXI provided the counties with an initial series of workshops in which international experts transferred knowledge about (evidence-based) best practices to improve performance on the HXI indicators. For each county, funding was conditional on their initiating and realizing planned achievements.

Respondents explicitly recognized the approach of HXI to be different from that of previous World Bank projects and from standing practices in the Chinese health sector.

> *In the previous projects, the counties were already told what to do and how to do it. They just followed orders. But for our project, HXI, they had to figure out themselves how to improve.*
>
> *(Respondent N2)*

The three case study counties welcomed HXI and had already earned recognition for prior achievements. They joined HXI aiming to resolve issues they had already identified prior to starting HXI. Mei Xian, for instance, had earned the distinction of being a 'national health county' for its achievements in (public) health services and had participated in a predecessor World Bank sponsored health sector reform project called Health VIII (HVIII). Within the public health component of HXI, Mei Xian was eager to start performance-based management, which they had already unsuccessfully tried to implement in 2006. Xi Xian was considered to have also performed well in HVIII, during which they developed their first ideas about jointly reforming hospital financing and performance.

> *"HVIII basically didn't provide a comprehensive intervention. But for the time when we had HXI, it was as if all the pieces were there."*
>
> *(respondent H1)*

Despite past performance recognitions, Jiulongpo was initially not selected for HXI. The county lobbied hard to be included, thereby causing their province to have six counties participating in HXI rather than the intended number of five.

Most HXI counties immediately ran into severe difficulties at the first step of selecting (international) best-practice interventions according to locally prioritized needs and tailoring them to their local context. Only six counties managed to complete this task within the first half year, as specified in the HXI project plan, and they were only able to do so after considerable guidance.

> *The first batch included 6 counties. At that time, we launched numerous meetings to help them at every step, such as how to write the proposals and how to write the activity plans. . .*
>
> (Respondent N3)

The remaining 34 out of 40 counties managed to propose a first plan with reform interventions by mid-2009, after learning from the innovation plans developed by the first six counties.

Several respondents reported that the difficulties continued after the initial intervention selection and adaptation phase and stretched over a period of two years. Some counties simply found it difficult to change their behaviours and to take initiatives and responsibility for their own reform implementation:

*Some counties in the 40 counties were not successful in the project because they still followed the old way of doing things. They would just wait for the instruction from the top level and follow what was done by other pioneer counties. Therefore, they lagged behind in this project.*

*(Respondent N6)*

The counties that completed implementations often discovered that their initially selected and adapted interventions failed to deliver the planned performance improvement. For instance, Xi Xian developed a case-based hospital payment system that was intended to reduce costs and treat patients according to standardized clinical pathways.

*At the very start, in 2009, we started to perform single case payment...but few patients met the requirements to be involved in the payment scheme.*

*(Respondent H4)*

For this intervention, the difficulties appeared to relate to a lack of collaboration from the medical staff, who were dissatisfied with the financial implications. It took sustained and collaborative efforts, involving all stakeholders, and several rounds of adaptations before the innovation developed into the 'ABC hospital payment system' [23, 24]. This ABC system distinguished three pathways per condition and included all intended patients (see S6 Appendix, Box 2).

A similar situation occurred in Mei Xian when they implemented a new performance-based budgeting and salary framework (see S6 Appendix, Box 1). Mei Xian needed several rounds of adaptation until they were satisfied with the results.

*"We made a lot of turns..."*

*(Respondent M1)*

Selecting and adapting interventions to effectively address the prioritized local needs required a learning process of several iterations, where the successful implementation of the first design appeared to be expected.

## Experts help it happen

Despite their initial struggles, Jiulongpo, Mei Xian and Xi Xian were among the first counties to report progress towards their performance targets [17, 18]. These accomplishments were achieved with extensive expert assistance. HXI provided generous access to experts to help counties and to promote the capacity building of provincial- and county-level staff. There were four 'levels' of experts involved: international experts, national experts, provincial experts and county experts.

The interactions with the international experts played a prominent role in the early stages of HXI. These experts introduced the counties to international best practices, evidence and scientific principles. The international experts played a minor role in the subsequent translation, or adaptation, of the international best practices to local, Chinese contexts. This role has been

taken on by national experts. Respondents from Jiulongpo, Mei Xian, and Xi Xian reported frequently and operationally involving national experts. Local project managers highlighted lists of expert contact information on the wall next to their desks and sequences of email traffic and messages. Those from Xi Xian explicitly mentioned having interacted with all national experts involved in HXI.

> *"Actually, I think [name national expert] and me, we keep in touch frequently. You can have a glimpse of my emails. We emailed very often because [name national expert] is very interested in this project; in fact, the frequent visits are directly related to the great attractiveness of the project. Because of this interest, now, we communicate in various ways. . . email, phone, many."*
>
> *(Respondent H1)*

The tight cooperation of the county staff, experts and, in some cases, the provincial teams extended into long-term collaboration with some of the national and provincial experts. The respondents speak of a 'network' in which they interactively communicated when needed.

## Top-down supervision and monitoring

The plans as initially developed by the participating counties, as well as subsequent yearly plans, went through an extensive assessment process involving several hierarchical levels. First, counties submitted their plans to provincial project management offices, where they received feedback from provincial health bureau staff and provincial experts. Counties had to revise their plans until they were approved by the provincial level project management office. Subsequently, the plans were submitted to the national office CPSM. The CPSM asked a panel of 14 national experts to review the proposals. Their feedback was presented to the provinces, which took care of processing the feedback, sometimes again involving the counties. After one or several rounds of feedback by the national experts, the plans were considered by the CPSM for approval and submission to the World Bank. If the World Bank was also approved, the plan was accepted, and funds were allocated.

As part of the plans, counties were requested to propose indicators tailored to their local situation, in addition to the baseline set of 22 indicators. This further enabled a bottom-up process in which counties specified their own targets. These self-developed indicators were subsequently used for monitoring purposes as well. Counties were required to report their progress every three months to provincial project management and to provide a written report to CPSM at least twice a year. Moreover, the CPSM organized two national supervision meetings and two joint supervision meetings (with the World Bank) yearly. This tight planning and control cycle was new to all Chinese parties involved and is seen as an important success factor. It enabled CPSM to be well informed and in control, even though each county developed their own interventions.

> *The other success factor is the real-time monitoring and the collection of information. As central-level decision makers, it is important to know opinions from every level of the project. If you don't know the exact information from different sides when making decisions, you will not feel confident about the decision.*
>
> *(Respondent N3)*

Especially in the early years of the project, monitoring took place on a detailed level. More so, as the reported progress on the indicators was used to trigger the release of funds and

thereby formed a pay for performance system. Hence counties were incentivized to perform according to plan. In addition, the HXI indicators were used by CPSM for benchmarking and to reward high-ranking counties.

At the same time, CPSM leadership recognized the limits set by the bottom-up approach for top-down control and enacted an enabling role, thereby stimulating the counties to learn and become more independent:

> . . ..the project counties and I are friends and I only give them guidance or support but no orders. I won't criticize them, and I think it makes no sense. I think the meetings are very effective.
>
> (Respondent N8).

Below, we further elaborate on the concurrent top-down and bottom-up mechanisms at play within HXI and the demands they placed on leadership.

### A new leadership style

County-level leaders were directly responsible for fostering a learning process and bottom-up reform tailored to local needs, while at the same time having to respond to formal top-down demands and deliver measurable performance improvement. It appears that the top-down forces created a context that gave legitimacy to promoting a bottom-up approach.

> "In China, if you want to get something new to be done, you start with the top-down measure, otherwise, no one would do it. You can say that if something is to be implemented, you must use this top-down method. . . .. So you have to get it done or you will lose your position. After the policy is in place, you also need the bottom-up approach to experiment and to pilot, so you can have more detailed operational guidelines and indicators".
>
> (Respondent H2)

While their position was particularly demanding when the bottom-up structures and processes lacked effectiveness, their pivotal position also empowered county-level leaders to play a crucial role. Many county-level respondents stressed the importance of county-level leadership as a reform success factor (see S6 Appendix, Box 3)

> Q: [. . .] Why is this district considered excellent?
>
> A: They have a very good leader, the director, who was able to identify issues.
>
> (Respondent J6)

When asked for specific leadership qualities, the respondents typically mentioned qualities associated with enabling bottom-up reform, such as role modelling and promoting equity, and they gave examples of servant leadership, including '*being able to stimulate the interest of subordinates and translate that into intrinsic motivation*' (Xi Xian), '*bringing people together to do a better job and facilitate better patient care*' (Mei Xian), and '*having the interest of subordinates at heart, both in job and in personal life, and stimulate teambuilding and solidarity among team members.*' (Jiulongpo).

This leadership style was also instrumental, as the HXI processes of adaptation and (incremental) improvements entailed involving '*stakeholders*', reaching '*consensus*', '*communication*', and other '*soft*' management competences.

*Health XI involves a lot of cooporations with other departments. So, whether the leader is good at the cooperation with other departments and whether the leader can win the trust or get the support from other departments is also important for this project.*

*(Respondent N6)*

## Autonomous learning

As mentioned above, the progress made during the first two years of HXI heavily relied on expert support. Each of the initially selected and adapted interventions came to exist after attending workshops and visits by international and national experts and the often detailed involvement in the design by national and provincial experts. In this first stage, most counties still basically ´followed the old ways', expecting to be instructed. County-level staff tended to seek instructions or apply reforms taught by experts and were less active in exploring their workings and developing appropriate adaptations.

Driven by failure to achieve the intended outcomes, they sought further expert assistance. Gradually, the bottom-up nature of HXI forced them into an explorative, cyclic, learning process. For example, to reward contribution towards achieving the performance targets, the county of Mei Xian needed a set of sub-indicators and sub-indicator weights to develop a performance-based management system (see S6 Appendix, Box 1). The first system design developed by provincial academic experts included 168 sub-indicators. However, this system appeared too complex to influence management and professional behaviours towards targeted performance improvement and was difficult to operate as well. As one of the county respondents stated:

"*We started with the university version of the plan,*

*and it did not go well because some data were too difficult to* collect."

*(Respondent M3)*

As county-level leadership held responsibility over the effectiveness of this reform intervention, it started to adapt the indicators together with county-level staff, involving experts when needed. Over the years 2010–2012, county-level staff reduced the number of indicators from 168 to 44, and performance improved.

Along the way, Mei Xian unintentionally learned how to autonomously solve reform design and implementation problems, depending only occasionally on experts. A similar development took place in Xi Xian while implementing and scaling up the ABC hospital payment system.

Jiulongpo even went so far as to state that HXI was not about implementing a specific intervention but rather about learning to intervene effectively after a baseline (problem) assessment.

*"The biggest success of the project is not about specific activities. More importantly, the project helped us to change the mindset to have new ways of doing things. We learned new evaluation tools. It is as if the project gave us a key that can open different doors. (. . .)*

*To put it another way, the national health reform points to a main destination, while HXI helped to find a roadmap to reach to destination. Whatever we do now, we refer to the methods that we learned from HXI. We first identify issues, solve the issue, evaluate it and assess the issue again and improve it until you are satisfied.*

*(Respondent J1)*

Each of the three counties we studied in-depth acquired the generic ability to autonomously address local health system performance problems and made doing so a habit, only involving their network of experts when required for specific issues.

## Discussion

China's almost 3,000 rural counties vary considerably on many determinants of health, health system design choices, and health needs of their populations. To accommodate such local differences, HXI piloted local health reforms tailored to local needs, following a hybrid bottom-up/top-down approach, thus departing from previously implemented top-down approaches [4, 9–11]. As argued in the introduction, this change in approach is ambitious, as it appears to match poorly with the culture and traditions of rural China [19–21]. In view of the effectiveness of HXI [4, 14–17] and our findings above, which indicate that the three studied pilot counties have in some respects advanced beyond the HXI objectives, we therefore set out to understand how the effective implementation of health reform interventions occurred in the hybrid bottom-up/top-down approach taken by HXI in the context of rural China.

As shown by our findings, HXI experienced difficulties right from the start. The country case study reports indicate that the three counties studied were ready for change [10, 11], and the presented results reveal that they have utilized the support and HXI project context as intended. We therefore attribute their difficulties to 1) county-level struggles to design radical changes and disruptive innovations that violated cultural norms of harmony, conformity and incremental/evolutionary progress [22, 23] and 2) difficulties to actively learn in autonomy, whereas the tradition of learning values instructions and relies on external expertise [21, 22]. Indeed, the respondents speak of the necessity to depart from 'the old way of doing things' and to a required 'change of mindset'. Before even learning new interventions, HXI required counties to unlearn their top-down behaviours. As only six out of 40 counties managed this initial task on schedule, while requiring much more support than anticipated, we find that learning bottom-up health reform has been a challenging innovation in itself in rural China.

Successful bottom-up reform requires a cyclic learning process of measuring, adjusting, implementing, measuring, etc. Counties only produced the targeted results after mastering this cyclic development process and going through several iterations. As the initial designs sometimes turned out to be ineffective, this required challenging the status quo, traditions, and (tacit) assumptions. It required higher-order learning skills. Instead of following instructions or expert advice, the counties learned how to identify problems and solve them through an iterative explorative process. Thus, they gradually acquired double-loop learning skills [32]. Moreover, these double-loop skills, which were not explicitly envisioned to develop within HXI, appear to have importantly determined the successes in the three studied counties.

Our findings indicate that these underlying determinants of implementation effectiveness strengthened the sustainability of the implemented reforms [10, 11, 33]. None of our findings indicated that counties had an interest in returning to 'the old way of doing things' or undoing their change of mindset. Whether the three case study counties will not only sustain their implemented innovations but also continue to practice their newly acquired problem-solving skills since HXI has ended and the support structures and process have disappeared is an interesting aspect of sustainability for future research.

HXI offered several enabling structures to overcome the aforementioned challenges. First, the generous provision of expert support was instrumental in designing the initial interventions and subsequently to iteratively resolving design shortcomings and implementation issues. The role of international experts was limited, apparently because the difference in tacit knowledge with county-level staff hampered the explicit knowledge transfer in the form of

international best practices [34]. However, counties eagerly relied on domestic experts who matched their traditional patterns of innovation [23]. Eventually, as county-level staff involved developed their capabilities, their relationships with domestic experts became more equal, and the consultations became more occasional.

Expert involvement was also part of the explicitly designed top-down assessment of reform plans and progress. It supported the tight top-down monitoring and supervision structure, which legitimized, pressured, stimulated, and enabled county-level reform. The formal and intense supervision structures and processes—which may be viewed to better match a top-down approach—appear to have been welcomed by all and to have enabled bottom-up adaptation. They matched the local cultures, and none of the respondents mentioned them as being counterproductive or paradoxical.

Taken together, the above findings suggest that the continuum of healthcare reform implementation approaches in which hybrid approaches reside—from bottom-up to top-down—has two dimensions: a content dimension and a procedural dimension. Within the content dimension, HXI relied on bottom-up practices such as diffusion, experimentation, adaption, autonomous and eventually double-loop learning [23, 32, 35], which are practices that match a 'let it happen' paradigm, as seen in the authoritative review of Greenhalgh et al. [10, 11] on the spread of health service innovations. To enable bottom-up innovation in the content dimensions, HXI relied on top-down practices in the procedural dimension. All county innovation activities were tightly supervised through an extensive planning and control framework, which included the release of funding. This top-down approach within the procedural dimensions matches a 'make it happen' perspective to disseminating innovation [11, 36].

Between 'let it happen' and 'make it happen', Greenhalgh et al. [11] position the 'help it happen' perspective, which emphasizes negotiation, influencing and enabling. Our findings confirm that such 'help it happen' practices played an important role in HXI. Both the HXI leadership itself and the experts intentionally played enabling roles, providing 'guidance and support' but 'no orders'. The relationships with county-level leadership were collaborative, and they were sometimes characterized as being 'friends'.

The role of county-level leaders appears to have been pivotal and essential. While influencing and negotiating with a complex stakeholder network ('help it happen'), they concurrently had to manage a novel bottom-up process on the content of innovation at the county level ('let it happen') and traditional top-down practices from higher levels ('make it happen'). They needed to handle the full complexity of HXI's ambitious design, which used much of the top-down to bottom-up continuum. Such a hybrid approach that explicitly blends procedural top-down elements into a bottom-up approach fits the rural Chinese context.

Hybrid approaches have been implemented and analysed in other healthcare contexts and found to be crucial in relation to healthcare quality and to be essential even in top-down systems [37, 38]. A two dimensional alternative to the one-dimensional bottom up–top down continuum has been proposed, in which "a vertical integration structure that can coordinate activity and manage potentially competing interests and motives" is complemented with a "horizontal or grassroots momentum" to capture these essentials [39, 40]. While our study echoes many of these findings, we hardly found evidence for an autonomous 'grassroot' momentum or an 'invisible' bottom up process in the counties participating in HXI [38, 39]. In fact, a main challenge within HXI has been to create bottom up momentum and it required considerable top-down efforts—as envisioned to some extent in the project design. Our findings thus confirm the relevance of hybrid approaches and indicate that context largely influences the delicate balance required to effectively complement bottom up mechanisms with top down mechanisms.

While our research design has provided an in-depth analysis in the three selected counties and at the HXI project level, it also has limitations. We have not been able to collect data in

counties which did not participate in HXI or in HXI counties that were reported to have had difficulties with realizing their objectives [9, 17]. Obviously, the latter enjoyed equal opportunities to benefit from top-down support structures and processes and had the same access to experts as the study counties. Is their performance difference then due to differences in readiness for change, county level leadership, or in learning capabilities? Or are there other factors at play that our research has failed to identify, such as complex local stakeholder relationships or selecting ill-suited reform interventions? Additionally, there may be differences regarding the sustainability of implemented reforms between best-practice counties and other counties. Furthermore, within the three counties, the (limited) number of respondents per county (ranging from 12 to 22) resulted from a priori data collection planning. Thus, even though we have covered best-practice counties from each of the three HXI themes and interviewed at all respondent levels, we cannot claim data saturation. As also suggested at mid-term and end term, future research on China's rural health reform can provide additional insights by adopting stronger research designs [14, 18].

## Conclusions

Local cultures and behaviours in rural China have posed additional challenges for hybrid health system reform approaches in comparison to such adoption in Western contexts or top-down approaches in rural China. These challenges surfaced, especially at the start of the ambitious and comprehensive reform project HXI. A change of mindset and persistence were needed to advance. Strong and supportive top down procedures and structures were set up to make the bottom-up reforms happen, further aided by enabling and supportive elements such as access to experts. County-level leadership is likely to have been pivotal in managing the complexity of this ambitious project design. The novel combination of top-down procedural elements with bottom-up elements to both let and help innovation happen fits the rural Chinese context well and confirms that the reform approach needs to be tailored to the context. Further robustly designed research in rural China and other contexts is called for.

## Supporting information

**S1 Appendix. Health XI indicators.**
(DOCX)

**S2 Appendix. Health XI counties and provinces.**
(DOCX)

**S3 Appendix. SRQR checklist.**
(DOCX)

**S4 Appendix. Interview protocol.**
(DOCX)

**S5 Appendix. HXI documents.**
(DOCX)

**S6 Appendix. County case study reports.**
(DOCX)

## Acknowledgments

We thank Yun Liu (YL) for her extensive help with translations and document analysis.

## Author Contributions

**Conceptualization:** Joris van de Klundert, Dirk de Korne, Jeroen van Wijngaarden.

**Data curation:** Joris van de Klundert, Dirk de Korne, Shasha Yuan, Fang Wang.

**Formal analysis:** Joris van de Klundert, Dirk de Korne, Jeroen van Wijngaarden.

**Funding acquisition:** Joris van de Klundert.

**Investigation:** Joris van de Klundert, Dirk de Korne, Shasha Yuan, Fang Wang.

**Methodology:** Joris van de Klundert, Dirk de Korne, Jeroen van Wijngaarden.

**Project administration:** Joris van de Klundert, Shasha Yuan, Fang Wang.

**Supervision:** Joris van de Klundert, Jeroen van Wijngaarden.

**Writing – original draft:** Joris van de Klundert, Dirk de Korne.

**Writing – review & editing:** Dirk de Korne, Shasha Yuan, Fang Wang, Jeroen van Wijngaarden.

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
