## [Decision Letter · Decision Letter 0]

21 Feb 2020

PONE-D-19-27227

From ‘make it happen’ to ‘help it happen’: a qualitative evaluation of health system innovation in rural China

PLOS ONE

Dear Prof. Klundert,

Thank you for submitting your manuscript to PLOS ONE. After careful consideration, we feel that it has merit but does not fully meet PLOS ONE’s publication criteria as it currently stands. Therefore, we invite you to submit a revised version of the manuscript that addresses the points raised during the review process.

The reviewers have provided thorough and detailed comments in an effort to strengthen the manuscript and study reporting. Please pay particular attention to the qualitative methods reporting and the integration of the theoretical framework throughout the paper. I look forward to receiving your revised manuscript. 

We would appreciate receiving your revised manuscript by Apr 06 2020 11:59PM. To enhance the reproducibility of your results, we recommend that if applicable you deposit your laboratory protocols in protocols.io, where a protocol can be assigned its own identifier (DOI) such that it can be cited independently in the future. For instructions see: http://journals.plos.org/plosone/s/submission-guidelines#loc-laboratory-protocols

We look forward to receiving your revised manuscript.

Kind regards,

Quinn Grundy, PhD, RN

Academic Editor

PLOS ONE

Journal Requirements:

2. When reporting the results of qualitative research, we suggest consulting the COREQ guidelines: http://intqhc.oxfordjournals.org/content/19/6/349. In this case, please consider including more information on the number of interviewers, their training and characteristics; how participants were selected; if a pilot study was tested; how data was coded; if bias issues were considered.

4. Please provide additional details regarding participant consent. In the ethics statement in the Methods and online submission information, please ensure that you have specified (1) whether consent was informed and (2) what type you obtained (for instance, written or verbal). If your study included minors, state whether you obtained consent from parents or guardians. If the need for consent was waived by the ethics committee, please include this information.

Yes, the research was funded by The World Bank, IDF Grant for Capacity Building of Evaluation of China's Health System Reform Pilot (No. TF013943). The originally collected data are owned by the Word Bank and can only be shared by the authors with consent of the World Bank.             

6. We note that you have indicated that data from this study are available upon request. PLOS only allows data to be available upon request if there are legal or ethical restrictions on sharing data publicly. For more information on unacceptable data access restrictions, please see http://journals.plos.org/plosone/s/data-availability#loc-unacceptable-data-access-restrictions.

7. Your ethics statement must appear in the Methods section of your manuscript. If your ethics statement is written in any section besides the Methods, please move it to the Methods section and delete it from any other section. Please also ensure that your ethics statement is included in your manuscript, as the ethics section of your online submission will not be published alongside your manuscript.

Reviewers' comments:

Reviewer's Responses to Questions

**Comments to the Author**

1. Is the manuscript technically sound, and do the data support the conclusions?

Reviewer #1: Partly

Reviewer #2: Yes

2. Has the statistical analysis been performed appropriately and rigorously? 

Reviewer #1: N/A

Reviewer #2: N/A

3. Have the authors made all data underlying the findings in their manuscript fully available?

Reviewer #1: Yes

Reviewer #2: Yes

4. Is the manuscript presented in an intelligible fashion and written in standard English?

Reviewer #1: Yes

Reviewer #2: Yes

5. Review Comments to the Author

Reviewer #1: 1.Background: please give more details of the intervention package of Health XI project in order to help the audience to better understand what the role and importance of "help it happen" approach in the project.

2. Research question: even though current literature showed the effectiveness of Health XI project somehow in China, it does not mean "bottom-up" reform happened in China. There might be many other factors/interventions contributing to the positive result of the Health XI project.

3. Materials and methods:

(1) Sampling strategy: the criteria for selection of research participants is not clear. Meanwhile, the authors did not say how to achieve information saturation. It seems the sampling is random. A minor question: in the text, it says 3 group interviews were conducted (Line 105). But in Table 1, there shows 4 focus group discussions.

(2) Interviewers: because qualitative researchers themselves are research tools, it is important to state in detail who did the interviews: do they have any experience in qualitative study? Have they both been trained for this study? How did the two interviewers speaking different language work together? Also, who did data analysis? How did the researchers do data analysis?

(3) Working language: it is very confusing. Regarding interviews, “these interviews were conducted in Chinese using certified interpreters who interpreted consecutively” (line 122-123). Does it mean the Chinese interviewer did the interviews in Chinese, and a translator did translation during the interview? Or how did you organize the interviews? It is not clear in the manuscript. Regarding data analysis, “the English texts of the interviews were transcribed using audio files of the original and translated interview content” (line 150-151). If the interviews are bilingual, how were they been transcribed and translated? This should be described more clearly in the paper.

(4) Ethics: because the study involved human participants, please list the name of the IRB or ethics committee and other information of the ethical approval. Otherwise, you should give sufficient information for why not obtaining ethical approval.

4. Results and discussion

(1) The trustworthiness of the results has been affected by the vagueness of the research method.

(2) The authors used a conceptual model of innovation in the analysis. But as a qualitative research, the original model should be deduced, deepened and finalized into a new model.

(3) The analysis of data is weak. The arguments do not sufficiently support the conclusions.

(4) A table with information (including such as case number/ age/ gender/ organization/ position et al.) of the participants should be presented in the paper.

In general, this is a project report rather than a scientific paper. The research design and methodology need to be improved. Meanwhile, the theory and analysis of governance in rural China and diffusion of changes are weak, which also need to be improved.

Reviewer #2: Summary of the research

The research team set out to explore the implementation of a bottom up process of reform in rural

China in a context of a traditionally hierarchical system. A ‘help it happen’ approach was adopted,

distinctive in its bottom up approach to reform implementation, as different from a traditional top

down approach to implementation.

Major comments:

Introduction

Page 3, line 42 – 43

Please include what these health system reforms hope to achieve. Please add at least five words

describing the overarching goal of health systems reform in China is. ….. “a series of national health

system reforms to achieve XXXXX and achieved considerable improvements in areas such as XXXXXXX

(1.2}

Other comments on introduction:

The introduction primarily speaks to literature on China, please provide a definition of ‘bottom up

approach’ with some international empirical examples of what typically happens in a bottom up

approaches to implementation, you might also want to clarify for the reader what you mean by top

down implementation. This will help the Reader to situate the work in broader literature on bottom

up implementation and why this work is unique. Not all readers will know what top down

reform/implementation versus bottom up reform/implementation is so it would be good to situate

the work – in all implementation of policies bottom up processes happen as implementers inevitably

use their discretion in implementation, the difference in your study is that bottom up processes were

intentional. Perhaps just one paragraph with a few key definitions and some empirical conclusions on

bottom up innovations in other contexts. You could add this to paragraph two.

Please define what ‘help it happen approach’ is. What are the key steps or principles? In your context

it seems that there are pre-set targets and there is a requirement that evidence-based innovations

are to be used when counties select innovations? Also a huge expert presence in this model. This

bottom up model is unique and the Reader should know this. The authors do note it is based on the

principle of ‘adaptation’ but it would be good to know a bit more about this process specifically.

Page 4, line 85

I struggled with the fact that the research question was a ‘what’ question - this is typically a

quantitative style of research question. This research, being qualitative in nature was more

exploratory trying to identify how and why success was achieved. I suggest rephrasing the question

to: How and in what contextual conditions did a bottom up reform process achieve success in three

rural Chinese districts?

This reflects a more appropriate exploratory study which this is, also given the importance of

understanding context when presenting qualitative work.

Page 4, line 90

Please make clear in the first paragraph to the Reader what the unit of analysis is in this paper. Are

you primarily exploring the routinisation and adoption of a bottom up process of designing local

innovations? Or are you primarily exploring the implementation of health reforms? Please link back

to you research question.

Good to be explicit to the Reader what is the primary object of analysis.

The process of trying to introduce bottom up processes (in a context of typically top down hierarchy)

is different from implementing reforms, even though they are linked. Good to also make link clear.

Page 4, line 97

Please explicitly include the selection criteria for the three districts. How and why were these three

selected from all the best practise counties.

Page 6, line 136 - 144

The application of theory and the use of conceptual frameworks is what allows qualitative research to

be transferable to other contexts, there is currently too little information on the key concepts and

theory applied. Please provide more detail on why the Greenhalgh f/work was used and what

specifically concepts were used in this study, please define the key concepts. Please explain why the

model by the World Bank was used and why you felt it relevant to merge the two (merging is fine but

the Reader should know why this specific two were merged). Please provide an explanation of what

is meant by “enriched with topics from organisational culture and learning”. What do these concepts

mean and how is it relevant to be used in this paper. While these concepts may be intuitive to some

from the social sciences, the PLOS audience goes beyond the social sciences. Also given that these

concepts are used in analysis this will further support deep understanding of the relevance of the

work.

In Appendix 4 a topic listing of words is insufficient, please include a definition of each of the concepts

alongside the word. The Reader should be able to see a definition of each concept while they are

reading to help understand the content.

Page 7, line 167

Please input a table that lists the 22 indicators, I felt as though I needed to know what this was to fully

understand the intentions of the reform. Tables typically don’t add to word count so this shouldn’t be

written in paragraph form but rather in a Table.

Page 7, 169

I am confused by the statement ‘self -formulated priorities’, given that they were asked to achieve a

specific set of 22 indicators which means that the targets/priorities were already set by national. I

think that what you are trying to say is that they were given the authority to develop bottom up

interventions to achieve the pre set targets? If you could clarify that would be good.

You also state in the next paragraph ‘prioritised improvement targets’.

Then on page 9, line 223 the authors write “self-set improvement targets”. Is this a selection from the

22 target indicators?

Only on page 10, line 256 is the Reader made aware that the counties could add targets. I am now

confused as to whether the targets/priorities spoken about earlier in the results section are referring

to these new targets or the old targets.

Page 7, line 183

Could you specify if counties were expected to only develop evidence-based interventions using

international best practice? Or were they also allowed to design innovations based on an

understanding of their own context and to think outside of the [evidence-based] box?

This is an important point to understand the nature of bottom up innovation in this paper. One

understanding of bottom up processes is where implementers craft innovations from their own

understandings of context and experiences. Your paper seems to include a particular nuance on top

of this.

Page 8, line 200

Spelling mistake on ‘implementations’

Page 8, line 218 – 219

I am not surprised by the many iterations and adaptations, this is the nature of bottom up processes

in implementation. What is interesting here is that implementers expected things to work out the first

time round – this tells me that perhaps there was a lack of communication on what bottom up reform

is and that in fact adaptation and learning is what should be expected.

Page 10

Suggestion: I really like that there is recognition in the paper that this model is in fact an interplay

between top down and bottom up reform/implementation. It might be useful for the authors to

acknowledge in the introduction that this model looks like a hybrid of top down / bottom up rather

than fully bottom up. The authors do not have to do this, but it might be good to qualify this upfront.

I think this links to me earlier point in the introduction section where the work needs to be situated

briefly in theory on what top down implementation is versus bottom up and the inevitability of both

being present.

Page 12. Line 326

I thought there were 22 indicators, and yet when I arrive at this point in the paper the Reader is advised

there are 166 indicators. I am therefore confused by the lack of clarity on the number of targets etc.

Please clarify and report so that the Reader is clear on which targets, set by whom, how many and in

which cycle of the process in the results section.

Page 13, line 366

Please explain in which ways you understand the three counties to have been ‘ready for change’.

Readiness for change is a critical part of reform adoption and implementation, thus the Reader would

benefit from this information. Please link this to your understanding to readiness for change as defined

by Greenhalgh.

Page 13, line 365 – 374

A suggestion: I assume also based on your results that the reason it was hard in the beginning is

because identifying innovations based on international literature is a very hard thing to do, local level

actors are often not trained to read empirical literature and then to rank interventions using evidence.

I think if you have any knowledge about the process of the experts engaging with the international

and sharing with the local actors this would be good to know to. I think it may be unfair to say only 6

out of 40 achieved this on schedule as it is a highly complex process of sorting through technical data.

As mentioned, in the introduction it would be good to know what the model of bottom up

implementation is – does it include evidence based innovations only or home grown also?

Page 13, line 380

Please list three examples of targets achieved. The reason I ask this is a simliar reason I asked for the

table earlier of the 22 indicators, while success is mentioned in the introduction I as the Reader do not

have a sense of what success was achieved, I realise that the success/benefits are published in another

paper but it would be nice to have some sense within the paper. I realise more information is given in

the Appendix. I am not asking for a major change in the paper in any way, just a sense here and there.

The goals of the reform are listed once only in the introduction but connections are not made again

explicitly to these goals in the paper, linking your findings in some way to these goals more explicitly

will help the Reader stay connected to the overall goals.

The conclusion

Things to consider: please see my comment above linked to page 13, line 365 – 374 and take into

consideration if my comment is relevant. I think important to note in the conclusion that while this

reform process was labelled as ‘bottom up” it was in fact an interplay between top down and bottom

up that drove success, which given the nature of expert support in this context is plausible.

6. PLOS authors have the option to publish the peer review history of their article (what does this mean?). If published, this will include your full peer review and any attached files.

Reviewer #1: No

Reviewer #2: No

---

## [Author Response · Author response to Decision Letter 0]

2 May 2020

please see cover letter and response to the reviewers for extensive responses

---

## [Decision Letter · Decision Letter 1]

26 Jun 2020

PONE-D-19-27227R1

‘Hybrid’ Top down Bottom up Health System Innovation in Rural China: a qualitative analysis

PLOS ONE

Dear Dr. Klundert,

Thank you for submitting your manuscript to PLOS ONE. After careful consideration, we feel that it has merit but does not fully meet PLOS ONE’s publication criteria as it currently stands. Therefore, we invite you to submit a revised version of the manuscript that addresses the points raised during the review process.

Thank you for your thorough response to the reviewers' comments, particularly in light of current global circumstances. We feel the manuscript is much strengthened, but have a few outstanding comments. Please address each of the reviewers' comments and the below. The reviewer has made a suggestion regarding the Discussion - I would appreciate your response or brief consideration of the literature.

The data availability and funding disclosure statements seem to be conflicting. In your statement about the involvement of the funder, please add that "the external evaluation of HXI was commissioned by the World Bank" and explain the relationship to data ownership. 

Minor comments:

- Abstract: Please address the use of the acronym Project HXI and spell out the first time. Please check the abstract for typos and grammar, including "hybrid'" (line 23), and several sentence fragments (lines 26-29)

- Introduction: be consistent in the use of double vs single quotation marks; line 106 and 108, what is "a large power difference?"

We look forward to receiving your revised manuscript.

Kind regards,

Quinn Grundy, PhD, RN

Academic Editor

PLOS ONE

Reviewers' comments:

Reviewer's Responses to Questions

**Comments to the Author**

1. If the authors have adequately addressed your comments raised in a previous round of review and you feel that this manuscript is now acceptable for publication, you may indicate that here to bypass the “Comments to the Author” section, enter your conflict of interest statement in the “Confidential to Editor” section, and submit your "Accept" recommendation.

Reviewer #2: (No Response)

2. Is the manuscript technically sound, and do the data support the conclusions?

Reviewer #2: (No Response)

3. Has the statistical analysis been performed appropriately and rigorously? 

Reviewer #2: (No Response)

4. Have the authors made all data underlying the findings in their manuscript fully available?

Reviewer #2: (No Response)

5. Is the manuscript presented in an intelligible fashion and written in standard English?

Reviewer #2: (No Response)

6. Review Comments to the Author

Reviewer #2: Introduction, page 3, line 48: Are all 800 million citizens in China rural? If not, please remove the word rural, it is confusing later when ‘rural’ is spoken about as a sample of the population rather than the whole population.

Line 67: IS HXI the acronym for China’s Rural Health Reform Project Health XI? If yes, please put in brackets at first use after the full word.

Line 90: Table 1 is referenced in this line as examples of interventions, but the information provided in Table 1 on page 7 does not match. Please correct.

Appendix 1: there is a spelling error in “The 22 Monitoring and Evaluation Indications of HXI”

Line 168: which ministry?

Line 172-174: I think there is a mix up with the Appendix numbering, the interview protocol is currently Appendix 5, not Appendix 4. Please check numbering of all appendices and correct where necessary.

Line 206-214: The bullet point style of numbering in this section looks out of place, perhaps write as follows: “These interviews were conducted as follows; (1) the researchers’ questions were always asked in English; (2) a certified interpreter translated the questions into Chinese and then asked them to the respondents; (3) …..

Line 241-257: In the data analysis section please provide at minimum one methodological reference that you consulted to develop your process of coding and analysis etc.

Line 251: you may want to also reference again the conceptual literature used (refs 19-23?) as you refer to these concepts again here (I realise it has been referenced earlier but it would be proper to reference again here as the concepts explicitly drawn on in the sentence.

Line 286: What is HVIII? This is a new acronym that is introduced but there is no full word description.

The discussion section: It is much improved, I value the analysis in the context of the Greenhalgh frameworks. But now that I can see the full body of work and can understand the research question in full, I have a suggestion (this is a suggestion) – you are meant to locate the discussion section in the context of other empirical literature, there are no other empirical examples referenced in your discussion section other than the work by Greenhalgh – you could identify other papers that apply the Greenhalgh concepts or that discuss a hybrid approach and compare/contrast with your findings to enrich your discussion section

7. PLOS authors have the option to publish the peer review history of their article (what does this mean?). If published, this will include your full peer review and any attached files.

Reviewer #2: No

---

## [Author Response · Author response to Decision Letter 1]

28 Aug 2020

See review response letter for all detailed responses

---

## [Editor Report · Decision Letter 2]

4 Sep 2020

‘Hybrid’ Top down Bottom up Health System Innovation in Rural China: a qualitative analysis

PONE-D-19-27227R2

Dear Dr. Klundert,

We’re pleased to inform you that your manuscript has been judged scientifically suitable for publication and will be formally accepted for publication once it meets all outstanding technical requirements.

Kind regards,

Quinn Grundy, PhD, RN

Academic Editor

PLOS ONE
---

## [Editor Report · Acceptance letter]

24 Sep 2020

PONE-D-19-27227R2 

‘Hybrid’ Top down Bottom up health system innovation in ruraL ChinA: a qualitative analysis 

Dear Dr. Klundert:

I'm pleased to inform you that your manuscript has been deemed suitable for publication in PLOS ONE. Congratulations! Your manuscript is now with our production department. 

Kind regards, 

on behalf of

Dr. Quinn Grundy 

Academic Editor

PLOS ONE